# `trajdata`: A Unified Interface to Multiple Human Trajectory Datasets

**Boris Ivanovic**[1]   **Guanyu Song**[2]   **Igor Gilitschenski**[2]   **Marco Pavone**[1,3]

[1]NVIDIA Research   [2]University of Toronto   [3]Stanford University

## Abstract

The field of trajectory forecasting has grown significantly in recent years, partially owing to the release of numerous large-scale, real-world human trajectory datasets for autonomous vehicles (AVs) and pedestrian motion tracking. While such datasets have been a boon for the community, they each use custom and unique data formats and APIs, making it cumbersome for researchers to train and evaluate methods across multiple datasets. To remedy this, we present `trajdata`: a unified interface to multiple human trajectory datasets. At its core, `trajdata` provides a simple, uniform, and efficient representation and API for trajectory and map data. As a demonstration of its capabilities, in this work we conduct a comprehensive empirical evaluation of existing trajectory datasets, providing users with a rich understanding of the data underpinning much of current pedestrian and AV motion forecasting research, and proposing suggestions for future datasets from these insights. `trajdata` is permissively licensed (Apache 2.0) and can be accessed online at https://github.com/NVlabs/trajdata.

## 1   Introduction

Research in trajectory forecasting (i.e., predicting where an agent will be in the future) has grown significantly in recent years, partially owing to the success of deep learning methods on the task [1]; availability of new large-scale, real-world datasets (see Fig. 1); and investment in its deployment within domains such as autonomous vehicles (AVs) [2, 3, 4, 5, 6, 7, 8, 9] and social robots [10, 11, 12].

In addition, recent dataset releases have held associated prediction challenges which have periodically benchmarked the field and spurned new developments [13, 14, 15, 16]. While this has been a boon for research progress, each dataset has a unique data format and development API, making it cumbersome for researchers to train and evaluate methods across multiple datasets. For instance, the recent Waymo Open Motion dataset employs binary TFRecords [17] which differ significantly from nuScenes' foreign-key format [18] and Woven Planet (Lyft) Level 5's compressed zarr files [19]. The variety of data formats has also hindered research on topics which either require or greatly benefit from multi-dataset comparisons, such as prediction model generalization (e.g., [20, 21]). To remedy this, we present `trajdata`: a unified interface to multiple human trajectory datasets.

**Contributions.** Our key contributions are threefold. First, we introduce a standard and simple data format for trajectory and map data, as well as an extensible API to access and transform such data for research use. Second, we conduct a comprehensive empirical evaluation of existing trajectory datasets, providing users with a richer understanding of the data underpinning much of pedestrian and AV motion forecasting research. Finally, we leverage insights from these analyses to provide suggestions for future dataset releases.

37th Conference on Neural Information Processing Systems (NeurIPS 2023) Track on Datasets and Benchmarks.

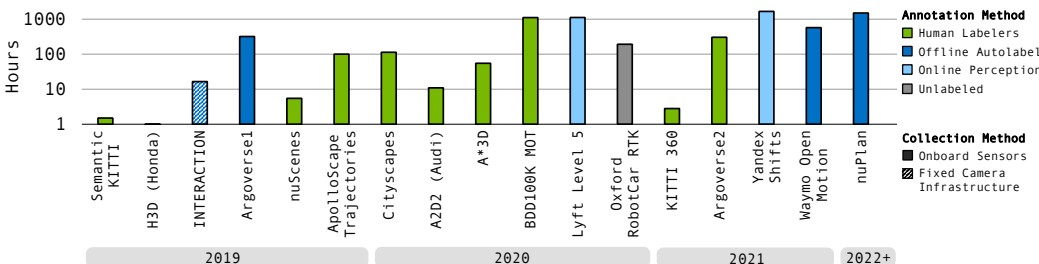

Figure 1: Recent datasets provide access to thousands of hours of autonomous driving data, albeit with different data formats and APIs, complicating the use of multiple datasets in research projects.

## 2 Related Work

**Human Trajectory Datasets.** Initial trajectory forecasting research employed video motion tracking datasets for benchmarking, primarily due to the availability of annotated agent positions over time. Of these, the ETH [22] and UCY [23] pedestrian datasets were among the most widely-used [1], containing a total of 1536 pedestrians and challenging behaviors such as couples walking together, groups crossing each other, and groups forming and dispersing. Soon after the successful application of deep learning models to pedestrian trajectory forecasting [24], and as data needs grew in autonomous driving research and industry, numerous large-scale datasets have emerged containing significantly more heterogeneous-agent interactive scenarios (e.g., between vehicles and pedestrians) in urban environments. Fig. 1 visualizes the scale, collection, and annotation strategy of such datasets, with a comprehensive review of earlier human motion datasets available in [1, 25]. In particular, the gradual shift from human annotation to autolabeling can be seen, with the recent large-scale Yandex Shifts [26], Waymo Open Motion [17], and nuPlan [27] datasets employing powerful autolabeling pipelines to accurately label sensor data collected by vehicle fleets at scale.

**Multi-Dataset Benchmarking.** While the increase in datasets and associated challenges has bolstered research, their unique formats increase the complexity of evaluating methods across datasets, complicating efforts to analyze, e.g., prediction model generalization. To address this issue for pedestrian motion data, OpenTraj [25] created dataloaders for different pedestrian motion datasets as part of its effort to evaluate and compare motion complexity across pedestrian datasets. More recently, TrajNet++ [28] and Atlas [29] present multi-dataset benchmarks to systematically evaluate human motion trajectory prediction algorithms in a unified framework. While these efforts have provided the community with multi-dataset benchmarks, they are primarily focused on pedestrian data. In contrast, `trajdata` tackles the standardization of both pedestrian *and* autonomous vehicle datasets, including additional data modalities such as maps.

## 3 `trajdata`: A Unified Interface to Multiple Human Trajectory Datasets

`trajdata` is a software package that efficiently compiles multiple disparate dataset formats into one canonical format, with an API to access and transform that data for use in downstream frameworks (e.g., PyTorch [30], which is natively supported). Currently, `trajdata` supports 8 diverse datasets, comprising 3,216 hours of data, 200+ million unique agents, and 10+ locations across 7 countries (see Table 1). To date, `trajdata` has been extensively used in research on trajectory forecasting [21], pedestrian [31] and vehicle [32, 33] simulation, and AV motion planning [34, 35].

### 3.1 Standardized Trajectory and Map Formats

**Trajectories.** For each dataset, `trajdata` extracts position, velocity, acceleration, heading, and extent (length, width, height) information for all agents in standard SI units (see Fig. 2). In order to support a variety of dataset formats, `trajdata` has minimal base data requirements: As long as agent positions (i.e., $x, y$ coordinates) are provided, all other dynamic information can be derived automatically. If entire dynamic quantities (e.g., velocity) are not captured in the original dataset, `trajdata` uses finite differences to compute derivatives by default. Further, missing data between

Table 1: Datasets currently supported by `trajdata`. More details can be found in the appendix.

| Dataset | Size | Locations | Maps? | Dataset | Size | Locations | Maps? |
|---------|------|-----------|-------|---------|------|-----------|-------|
| ETH [22] | 0.4h | 2 | No | INTERACTION [39] | 16.5h | 4 | Yes |
| UCY [23] | 0.3h | 2 | No | Lyft Level 5 [19] | 1118h | 1 | Yes |
| SDD [40] | 5h | 1 | No | Waymo Open [17] | 570h | 6 | Yes |
| nuScenes [18] | 5.5h | 2 | Yes | nuPlan [27] | 1500h | 4 | Yes |

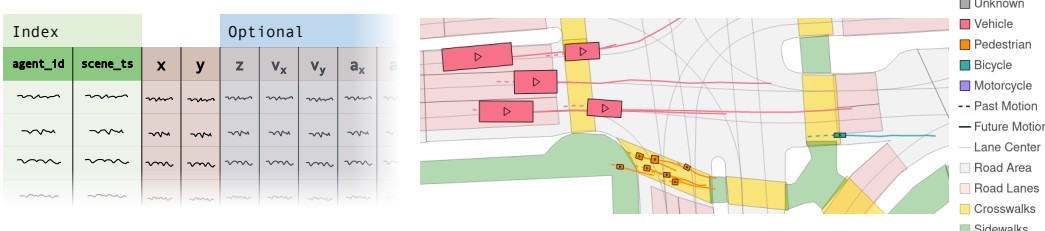

Figure 2: **Left:** `trajdata` adopts a tabular representation for trajectory data, leveraging advanced indexing to satisfy user data queries. **Right:** Agent trajectories from the nuScenes [18] dataset visualized on the scene's `VectorMap`, containing all of `trajdata`'s core map elements.

timesteps is imputed via linear interpolation. `trajdata` internally represents and stores trajectory data as tabular data frames, allowing for advanced indexing and data grouping depending on user queries and the use of efficient open-source tabular data storage frameworks such as Apache Arrow [36]. Note that each of these default choices (finite differences, linear interpolation, and tabular data frames) can be changed by the end user.

**Maps.** To retain the most information from high-definition (HD) dataset maps, `trajdata` adopts a polyline representation for map data. This choice matches the vast majority of modern trajectory datasets which provide vector map data and makes them immediately compatible with our format. Currently, there are four core map elements: `RoadLane`, `RoadArea`, `PedCrosswalk`, and `PedWalkway`. As illustrated in Fig. 2, a `RoadLane` represents a driveable road lane with a centerline and optional left and right boundaries. A `RoadArea` represents other driveable areas of roads which are not part of lanes, e.g., parking lots or shoulders. A `PedCrosswalk` denotes a marked area where pedestrians can cross the road. Finally, a `PedWalkway` marks sidewalks adjacent to roads. Of these, only `RoadLane` elements are required to be extracted, other elements are optional (they are absent in some datasets). Our map format additionally supports lane connectivity information in the form of left/right adjacent lanes (i.e., lanes accessible by left/right lane changes) and successor/predecessor lanes (i.e., lanes that continue from / lead into the current lane following the road direction).

Each map element is designed to be compatible with popular computational geometry packages, such as Shapely [37], enabling efficient set-theoretic queries to calculate, e.g., road boundary violations. By default, `trajdata` serializes map data using Google protocol buffers [38], and, in particular, only stores neighboring position *differences* for efficiency, similar to the implementation used in [19]. Dynamic traffic light information is also supported, and `trajdata` makes use of a separate data frame to link the traffic signal shown per timestep with the lane ID being controlled.

## 3.2 Core `trajdata` Functionalities

**Multi-dataset training and evaluation.** One of `trajdata`'s core functionalities[1] is aggregating data from multiple datasets in a `UnifiedDataset` object (a PyTorch [30] `Dataset` subclass by default).

```
1 from trajdata import UnifiedDataset
2 dataset = UnifiedDataset(
3   desired_data=["nusc_mini-boston", "sdd-train"], desired_dt=0.1,
4   centric="agent", history_sec=(1.0, 3.0), future_sec=(4.0, 4.0)
5 ) # These settings were used to create Figure 2.
```

---

[1] Detailed demonstrations of `trajdata`'s capabilities can be found in our repository's `examples/` folder.

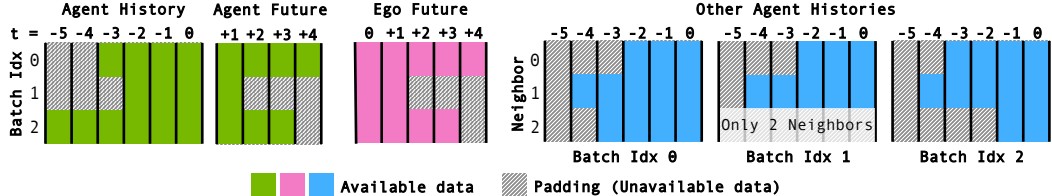

Figure 3: `trajdata` can provide agent-centric (or scene-centric) batches of trajectory data for model training and evaluation in associated `AgentBatch` (or `SceneBatch`) objects. The indexing and padding strategy of a few core `AgentBatch` tensors are visualized here.

The example above creates a dataset that provides agent-centric data batches (i.e., each batch element contains data for one agent at one timestep, see Fig. 3) sourced from only Boston in the nuScenes mini dataset (`"nusc_mini-boston"`) as well as the Stanford Drone Dataset's entire training split (`"sdd-train"`), with time upsampling ensuring all data is at 10Hz (`desired_dt=0.1`). `history_sec=(1.0, 3.0)` specifies that the predicted agent's trajectory must have at least $1.0s$ of history available, with padding for any missing data up to $3.0s$ (see Fig. 3). Similarly, `future_sec=(4.0, 4.0)` requires that the predicted agent's trajectory have $4.0s$ of future available.

`trajdata` provides many other capabilities in addition to the above, including scene-centric batches (i.e., data for all agents in a scene at the same timestep), semantic search (e.g., nuScenes [18] provides text descriptions for each scene), agent filtering (e.g., only vehicles), coordinate frame standardization (i.e., making trajectories relative to the predicted agent's frame at the current timestep), map rasterization (e.g., if encoding scene context with a convolutional architecture), data augmentations (e.g., additive Gaussian noise to past trajctories), and general data transforms via custom functions.

**Map API.** `trajdata`'s standardized vector map object is `VectorMap`. In addition to providing access to individual map elements (e.g., lanes, sidewalks), it also leverages precomputed spatial indices to make nearest neighbor queries very efficient.

```
1 from trajdata import MapAPI, VectorMap
2 vec_map: VectorMap = MapAPI(<=>).get_map("nusc_mini:boston-seaport")
3 lane = vec_map.get_closest_lane(np.array([50.0, 100.0, 0.0]))
```

In the example above, the polyline map of Boston's seaport neighborhood (from nuScenes [18]) is loaded from the user's `trajdata` cache (its path would be specified instead of `<=>`) and queried for the closest `RoadLane` to a given $x, y, z$ position.

**Simulation Interface.** `trajdata` also provides a simulation interface that enables users to initialize a scene from real-world data and simulate agents from a specific timestep onwards. Simulated agent motion is recorded by `trajdata` and can be analyzed with a library of evaluation metrics (e.g., collision and offroad rates, statistical differences to real-world data distributions) or exported to disk. This functionality was extensively used to benchmark learning-based traffic models in [32, 33].

```
1 from trajdata.simulation import SimulationScene
2 sim_scene = SimulationScene(<=>) # Specify initial scene to use.
3 obs = sim_scene.reset() # Initialized from real agent states in data.
4 for t in range(10): # Simulating 10 timesteps in this example.
5     new_state_dict = ... # Compute the new state of sim agents.
6     obs = sim_scene.step(new_state_dict)
```

In this example, a `SimulationScene` is initialized from a scene in an existing dataset (specified with the `<=>` arguments), after which it can be accessed similarly to an OpenAI Gym [41] reinforcement learning environment, using methods like `reset` and `step`.

## 4   Dataset Comparisons and Analyses

In this section, we leverage `trajdata`'s standardized trajectory and map representations to directly compare many popular AV and pedestrian trajectory datasets along a variety of metrics. Our goal is to provide a deeper understanding of the datasets underpinning much of human motion research by analyzing their data distributions, motion complexity, and annotation quality.

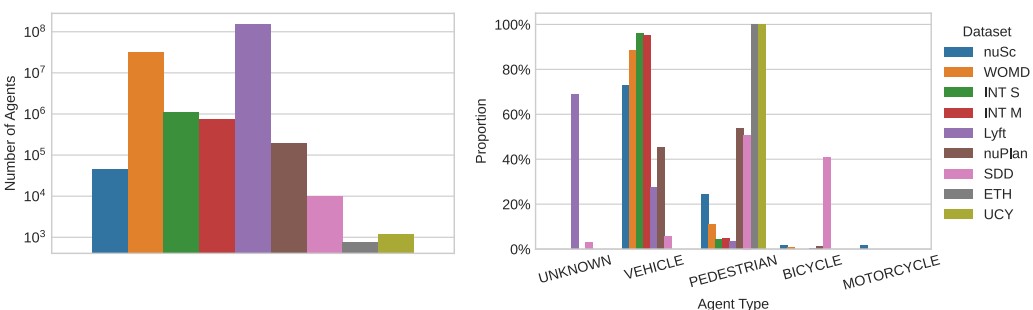

Figure 4: **Left:** Number of unique agents per dataset. **Right:** Distribution of agent types per dataset.

Note that we only analyze dataset training and validation splits, since these are the splits predominantly used by methods for development. We explicitly do not analyze test splits since they are either not available publicly or because doing so may harm existing benchmark validity. Further, while `trajdata` supports data frequency up- and down-scaling via interpolation and down-sampling, all of the following analyses were conducted in each dataset's native data resolution. All analyses were performed using the latest version of `trajdata` at the time of writing (v1.3.2) on a desktop computer with 64 GB of RAM and an AMD Ryzen Threadripper PRO 3975WX 32-core CPU. For larger datasets, an NVIDIA DGX-1 server with 400 GB of RAM and 64 CPU cores was used.

## 4.1 Agent Distributions

**Population.** To build a fundamental understanding of the considered datasets, we first analyze and compare agent populations. Fig. 4 visualizes overall agent counts and proportions per dataset. As can be expected, modern large-scale AV datasets such as Waymo [17] and Lyft Level 5 [19] contain multiple orders of magnitude more agents than earlier pedestrian datasets SDD [40], ETH [22], or UCY [23]. However, as we will show later, pedestrian datasets still provide value in terms of agent diversity, density, and motion complexity in popular social robotics settings such as college campuses.

As can be seen in Fig. 4 (right), the vast majority of agents in AV datasets are vehicles or pedestrians, with the exception of Lyft Level 5 [19] where 71.8% of agents have unknown types. In contrast, bicycles (a relatively niche category in many datasets) account for 41% of all agents in SDD [40] (indeed, biking is a popular method of transportation around Stanford's large campus). Such imbalances in agent populations are indicative of real-world distributions, e.g., motorcycles make up only 3.5% of vehicles in the USA [42], similar to their proportion in nuScenes [18] (1.6%).

**Density and Observation Duration.** In addition to which agent types are captured in scenes, the amount and density of agents can be an important desiderata (e.g., for research on crowd behavior) or computational consideration (e.g., for methods whose runtime scales with the number of agents). Fig. 5 visualizes the distribution of the number of agents observed per scene per timestep (left), as well as the *maximum* number of simultaneous agents per scene (right). As can be seen, urban scenarios captured in modern AV datasets frequently contain 100+ detected agents (with a long tail extending to 250+ agents). In this respect, ETH [22], UCY [23], and INTERACTION [39] are limited by their fixed-camera and drone-based data-collection strategies compared to the comprehensive on-vehicle sensors used in nuScenes [18], Waymo [17], Lyft [19], and nuPlan [27]. However, while ETH [22], UCY [23], and INTERACTION [39] do not contain as many agents, they consistently provide the highest-density scenes (see Fig. 6), especially for pedestrians and bicycles. We compute agent density by dividing the number of agents in a scene by their overall bounding rectangle area, as in [25].

Each dataset supported by `trajdata` adopts different scenario lengths and corresponding agent observation durations. As can be seen in Fig. 7, AV datasets are comprised of scenarios with lengths ranging from 4s in INTERACTION [39] to 25s in Lyft Level 5 [19]. The peaks at the right of each AV dataset duration distribution are caused by the always-present ego-vehicle (for Vehicles) as well as other agents detected throughout the scene (common in steady traffic, parking lots, or at an intersection with stopped traffic and pedestrians waiting to cross). One can also see that Lyft Level 5 [19] agent detections are much shorter-lived compared to other AV datasets' relatively uniform distributions (Waymo [17], nuScenes [18], and nuPlan [27]). This could be caused by Lyft's

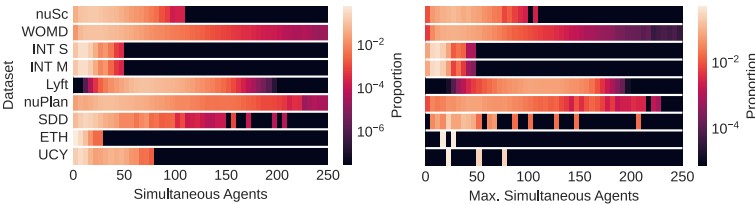

Figure 5: **Left:** Number of agents present per timestamp and scene. **Right:** Maximum number of agents present at the same time per scene.

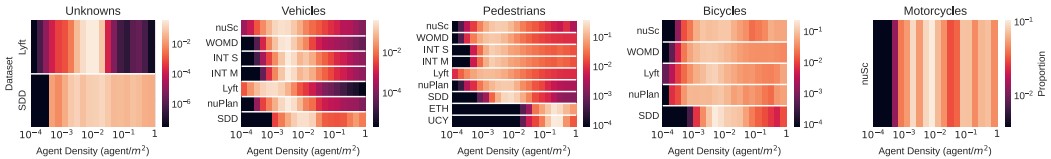

Figure 6: Agent density per timestep and scene.

annotations being collected from an onboard perception system [19] (which are affected by noise and occlusions) vs human annotators [18] or autolabeling [27, 17] which can leverage data from past and future timesteps be more robust to such errors. We conduct additional comparisons between data collection methodologies in Section 4.3.

**Ego-Agent Distances.** When developing AV perception systems, an important consideration is the sensor range(s) necessary to facilitate the desired prediction and planning horizons as well as provide advanced warning of critical situations (e.g., stopped traffic on a highway). In Fig. 8, we compare the distribution of ego-agent distances and find that, while nuScenes [18] and Lyft Level 5 [19] have long-tailed distributions extending past $200m$, Waymo [17] and nuPlan [27] appear to have artificial cut-offs at $75$-$80m$, potentially to maintain data quality by avoiding poor data from distant agents. However, it would be more useful to maintain distant detections and add uncertainty outputs from the autolabeler to support uncertain long-range detection research in addition to improving autolabeling.

**Mapped Areas.** HD maps are a core component of many AV datasets, frequently leveraged in trajectory forecasting and motion planning research to provide scene context and geometric lane information (e.g., for global search-based planning and trajectory optimization). Current AV dataset maps are very large (see Table 2 in the appendix) and comprehensive, spanning multiple neighborhoods in different cities. However, not all HD maps are created equal, commonly differing along three axes: Area completeness, lane definitions, and traffic lights. While most AV datasets provide complete HD maps of neighborhoods, Waymo [17] differs by only providing local map crops per scenario without a common reference frame across scenarios[2]. This also significantly increases the storage requirements of Waymo [17] maps compared to other datasets.

Lane definitions can also differ significantly between datasets, with intersections being a notable differentiator. For instance, the nuScenes dataset [18] does not annotate intersections fully, opting for only lane centerlines without associated edges (Fig. 2 shows an example). Lyft Level 5 [19] and nuPlan [27] both include full lane center and edge information for all possible motion paths through an intersection. Waymo [17] maps are unique in that they provide full lane center and boundary information, but there are many gaps in the associations between lane centerlines and boundaries, making it difficult to construct lane edge polylines or lane area polygons[3]. As a result, we exclude Waymo maps from map-based analyses in this work.

## 4.2 Motion Complexity

Measuring the complexity of driving scenarios is an important open problem in the AV domain, with a variety of proposed approaches ranging from heuristic methods [25] to powerful conditional behavior

---

[2]See https://github.com/waymo-research/waymo-open-dataset/issues/394 for visualizations.

[3]See https://github.com/waymo-research/waymo-open-dataset/issues/389 for visualizations.

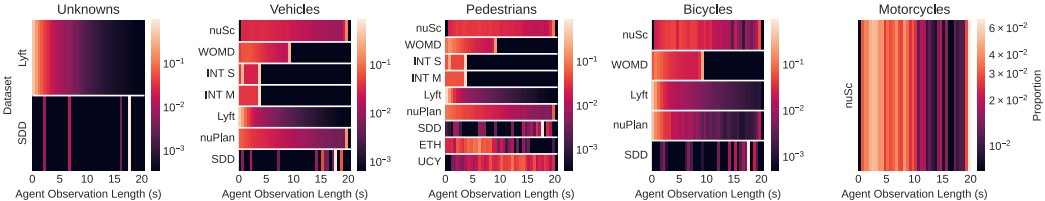

Figure 7: Distributions of the length of time agents are observed in each scene.

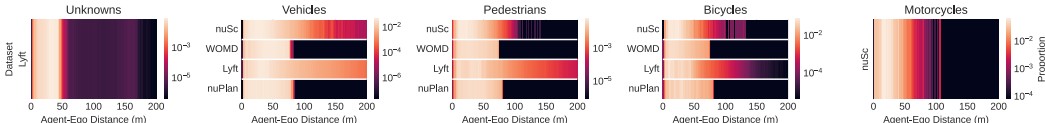

Figure 8: Distribution of distances between agents and data-collecting ego-vehicle in AV datasets.

prediction models [43]. To avoid potential biases in analyzing datasets with an externally-trained model, we employ simple and interpretable heuristics similar to [25].

**Motion Diversity.** We first analyze distributions of dynamic agent quantities (e.g., speed, acceleration, jerk). As can be seen in Fig. 9, the majority of speed distributions have high peaks at zero (no motion). This is corroborated by Table 3 in the appendix, which shows that a significant portion of agents are stationary in many datasets, especially for nuScenes [18] (17.5%) and Waymo [17] (53.6%). After the initial peak, agent speed distributions drop sharply to a roughly uniform plateau (up to $20m/s$ for vehicles) before dropping completely around $30m/s$ (a common highway speed around the world).

While SDD [40] and INTERACTION [39] have sensible vehicle speeds, their pedestrian speeds can be too high. Such high speeds may be caused by annotations near the edge of drone camera view or by rectification artifacts near the image border. Additionally, the very long-tailed distribution of Lyft [19]) and Waymo [17]) vehicle, pedestrian, and bicycle speeds (exceeding $60m/s$) show a remaining area of improvement for state-of-the-art AV perception systems and autolabeling pipelines. Comparisons of acceleration and jerk can be found in the appendix. Overall, from dynamic quantities alone, Waymo [17]) and Lyft [19] provide the most diversity in agent motion. If such long-tailed data is undesirable, the INTERACTION [39] dataset provides the most realistic set of vehicle speeds.

**Trajectory Nonlinearity.** To analyze the spatial diversity of agent trajectories, we first compare each agent's heading to their initial timestep. As can be seen in Fig. 10, and reiterating earlier analyses, the vast majority of human movement is straight and linear ($\Delta h = 0$). Moving away from the center, we also see repeated symmetric peaks at $\pm\frac{\pi}{2}$ (capturing left and right turns) and $\pm k\pi$ in some datasets. One possible reason for these periodic peaks in the distribution is an artifact of the autolabeling methods used in the datasets (since only datasets that autolabel sensor data are affected), another is that their respective scene geometries contain more roundabouts, cul-de-sacs, and repeated turns than other datasets (more detailed heading distributions can be found in the appendix). We can also see that pedestrians' distributions are more uniform as they do not have to adhere to rigid road geometry.

**Path Efficiency.** Lastly, we also measure agent path efficiencies, defined as the ratio of the distance between trajectory endpoints to the trajectory length [25]. Intuitively, the closer to $100\%$, the closer the trajectory is to a straight line. As can be seen in Fig. 15 in the appendix, most path efficiency distributions are uniformly distributed, with peaks near $100\%$, echoing earlier straight-line findings. However, the INTERACTION [39] dataset is an outlier in that its agent trajectories are predominantly straight lines with much less curved motion than other AV and pedestrian datasets.

## 4.3 Annotation Quality

While analyzing datasets' true annotation accuracy would be best, neither we nor the original data annotators have access to the underlying real-world ground truth. As a proxy, we instead analyze the *self-consistency* of annotations in the form of incidence rates of collisions between agents, off-road driving, and uncomfortable high-acceleration events (using $0.4g$ as a standard threshold [44, 45]).



Figure 9: Agent speed distributions per dataset and agent type.

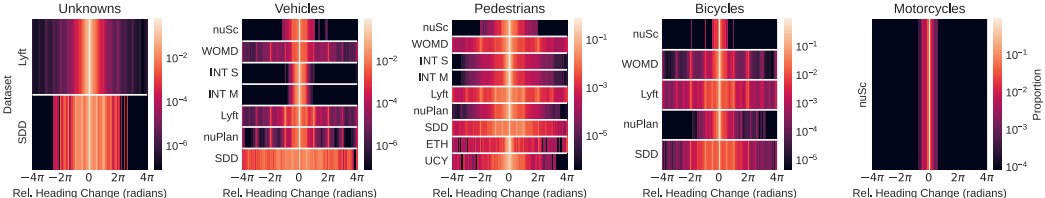

Figure 10: Changes in heading relative to an agent's first timestep.

Virtually all observed agent data is free of collisions and off-road driving, save for rare one-offs (e.g., the INTERACTION dataset contains a minor car accident [39]). We denote bounding box intersections between agents as collisions, and agent center-of-mass exiting the road boundary as off-road driving. Collisions typically indicate errors in bounding box annotations, whereas off-road driving can indicate erroneous bounding box dimensions, missing map coverage, or harsh driving that, e.g., cuts corners during a right turn.

As can be seen in Fig. 11 (left), most vehicles in datasets experience collision rates below 5%. Of particular note is the fact that state-of-the-art autolabeling systems (e.g., used in Waymo [17]) are nearly matching the accuracy of human annotations (e.g., used in nuscenes [18]) in terms of resulting collision rates. However, detecting agents from a near-ground perspective (even with 3D LiDAR) is a very challenging task, and current performance still lags behind high altitude viewpoints. In particular, The INTERACTION [39] dataset achieves orders of magnitude lower vehicle collision, off-road, and harsh acceleration rates owing to its drone-based data collection strategy. In theory, SDD [40] should enjoy a similar advantage, but it only provides axis-aligned bounding box annotations (which overestimate agent extents) and Stanford's college campus contains much more interactive agents than other urban environments. More generally, the notion of bounding box intersections as collisions does not transfer exactly to pedestrians as they can enter/exit cars and walk in close groups, and further study is needed to robustly distinguish between errant motion and normal interactive motion.

In Fig. 11 (middle), we find that vehicles in general experience very few ($< 1\%$) harsh acceleration events, with Waymo [17], Lyft [19], and nuScenes [18] all having the highest incidence, commensurate with their earlier-discussed long-tail acceleration distributions. Lastly, we find in Fig. 11 (right) that the INTERACTION [39] and nuPlan [27] agent annotations are well-aligned onto their maps, whereas nuScenes [18] suffers from poor map coverage away from main roads (there are many annotated parked cars next to the main road) and Lyft [19] suffers from high false positive detections next to the main road (the majority of which take the Unknown class).

## 5 Conclusions and Recommendations

The recent releases of large-scale human trajectory datasets have significantly accelerated the field of AV research. However, their unique data formats and custom developer APIs have complicated multi-dataset research efforts (e.g., [20, 21]). In this work, we present `trajdata`, a unified trajectory data loader that aims to harmonize data formats, standardize data access APIs, and simplify the process of using multiple AV datasets within the AV research community with a simple, uniform, and efficient data representation and development API. We used `trajdata` to comprehensively compare existing trajectory datasets, finding that, in terms of annotation self-consistency, drone-based data collection methods yield significantly more accurate birds-eye view bounding box annotations than even state-of-the-art AV perception stacks with LiDAR (albeit with much less spatial coverage), modern

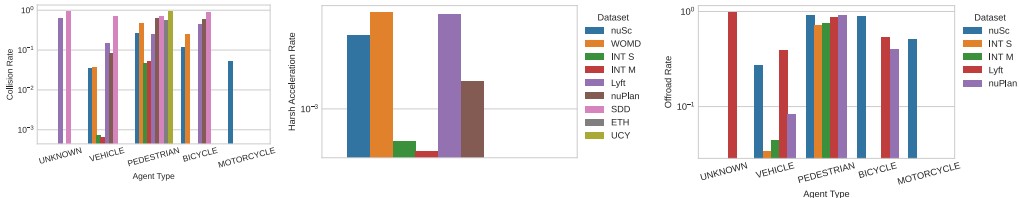

Figure 11: Self-consistency failure rates per dataset and agent type, in the form of collision (**left**), high vehicle acceleration (**middle**), and off-road (**right**) rates.

autolabeling pipelines are nearing human annotation performance, and smaller-scale pedestrian datasets can still be useful for investigations requiring high-agent-density scenarios.

As concrete recommendations, we saw that some datasets artificially limit the distance agents are autolabeled. Instead, it would be more useful to the long-range detection community to remove such restrictions, but add autolabeler-output uncertainties to long-range detections, supporting uncertain perception research along the way. Further, incorporating explicit self-consistency checks within autolabeling pipelines and catching, e.g., collisions, prior to release can both improve the autolabeling method as well as the resulting data labels.

More broadly, providing researchers with access to more data comprised of various agent types from diverse geographies should help in modeling rare agent types and behaviors, in addition to aiding in the generalization of methods to multiple geographies. However, as we have seen in prior sections, there is an *overwhelming* bias towards straight line driving, and one capability missing from `trajdata` is the ability to (re)balance data on a semantic (behavioral) level. Finally, even if lower-level trajectory classes (e.g., driving straight, turning left/right, slowing down, speeding up, etc) are balanced, an important higher-level consideration during original dataset curation time is to ensure that AV datasets explore *all* geographic regions within an environment, and not only those of certain socioeconomic statuses or transportation access.

Future work will address the current limitations of `trajdata` (e.g., expanding the number of supported datasets and new capabilities such as geometric map element associations to support Waymo-like map formats [17]). Further, incorporating sensor data would also enable perception research as well as joint perception-prediction-planning research, an exciting emerging AV research field.

## Acknowledgments and Disclosure of Funding

We thank all past and present members of the NVIDIA Autonomous Vehicle Research Group for their code contributions to `trajdata` and feedback after using it in projects. We additionally thank Leon De Andrade, Alex Naumann, and Stepan Konev for their contributions to `trajdata` on GitHub.

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
