# OpenReview forum: "trajdata: A Unified Interface to Multiple Human Trajectory Datasets"
_NeurIPS.cc/2023/Track/Datasets_and_Benchmarks — NeurIPS 2023 Datasets and Benchmarks Poster_

### Official Review · Reviewer_fsaD · 2023-07-19
**Review for trajdata**

**Rating:** 7
**Confidence:** 5
**Clarity:** The paper is generally organized well…

**Strengths:**

The introduction of Trajdata will make the researchers who work on trajectory forecasting tasks focus more on the algorithm/system design rather than dataset processing, given the following reasons:
1. It standardizes the trajectory data representation and the HD map, which used to be very different for different datasets.
2. The functions it supports are necessary and valuable. For example, it promotes multi-dataset benchmarking and agent-centric batch & scene-centric batch generation, which all highly align with the current research interests of the community.
3. The simulation interface is also well-motivated since data shortage is always one of the biggest problems in the trajectory forecasting community.

Meanwhile, the analysis of the existing datasets is meaningful. It provides in-depth statistic analysis from several perspectives, including the diversity of the agent types, the diversity of the trajecotires, and the agent density. This reviewer can tell the authors are very familiar with the topic and understand what information is that the researchers need when designing the systems.


**Additional Feedback:**

N/A

**Correctness:**

The claims are technically correct in the submission. Although the submission does not contain a new dataset, the way it standardizes the trajectory data and maps information from existing datasets seems reasonable.

**Documentation:**

The documentation on the GitHub webpage introduces how to use the interface and other necessary information.

**Ethics:**

Since the trajdata only includes the existing public datasets, the ethical concern may be very minor for this work.

**Limitations:**

In this submission, the authors have not shown the experiment results of existing methods testing on the proposed trajdata interface. Since trajdata involves some modification of the original data formats from the existing datasets, it would be necessary to make sure the validity of those modifications/transformations by testing existing methods on the modified data.

**Opportunities For Improvement:**

In L300-L302, the authors mentioned an imbalance of the motion types in the existing datasets. When evaluating the methods, researchers may want to specifically see how the system works in more difficult cases rather than the original validation set which *is overwhelmingly biased towards straight-line* driving with a constant speed. Therefore, it would be ideal to facilitate this need by letting users select data splits for testing/training given the statistical description of the scenes. For example, users can follow the authors' analysis in Figure 9 & Figure 10.


**Relation To Prior Work:**

Yes. It discussed the previous works in Sec. 2.

**Summary And Contributions:**

This paper proposed a unified interface, the trajdata, for trajectory forecasting datasets. It includes almost all the popular datasets in the field, such as SDD, UCF/ETH, Waymo, and Lyft. It standardizes the trajectory data samples from different datasets as well as the HD map information. Meanwhile, the paper conducts an analysis of the existing trajectory forecasting datasets to facilitate future research and the datasets release.

---

> ### Author Response · Authors · 2023-08-11
> **Rebuttal for Reviewer fsaD**
>
> Thank you for your thoughtful review and suggestions! We address detailed review comments point-by-point below.
>
> > it would be ideal to facilitate this need by letting users select data splits for testing/training given the statistical description of the scenes
>
> This is a great suggestion! trajdata has a few semantic scene selection capabilities implemented, but these can certainly be expanded, and we agree that enabling users to define their preferred data selection distributions is a flexible, general way to accomplish this. We will look into this, thank you!
>
> > the authors have not shown the experiment results of existing methods testing on the proposed trajdata interface. Since trajdata involves some modification of the original data formats from the existing datasets
>
> While we do not have such comparisons in the paper, other researchers using trajdata have validated that their models perform the same before and after switching their data loading framework. For example, another researcher on our team has verified that [AgentFormer](https://github.com/Khrylx/AgentFormer) matches its original performance after switching to trajdata.
>
> Aside from direct model comparisons, trajdata does not modify the original data (i.e., the raw trajectories and maps) if the user does not specify any options that modify those; all data transformations (e.g., standardization to the ego-vehicle’s current state) can be disabled via boolean arguments to the `UnifiedDataset` constructor.

---

### Official Review · Reviewer_8txH · 2023-07-21
**I personally like the unified interface idea, and we can consider accept it.**

**Rating:** 7
**Confidence:** 4
**Clarity:** It seems that the paper is basically …

**Strengths:**

Significance of the Contribution.The unified data loader "trajdata" is a significant contribution as it simplifies and standardizes the process of using and comparing multiple AV datasets. This tool can expedite research by eliminating the complexities associated with managing and analyzing datasets in different formats and using different APIs.
Relevance to the Broader Research Community. The research is highly relevant to the autonomous vehicle (AV) research community, as it offers a comprehensive comparison of existing trajectory datasets. Such an analysis aids in identifying strengths and weaknesses of different data collection and annotation methods and can guide future data collection efforts.
Quality of the Research.The research is thorough, offering a comprehensive analysis of various datasets, and providing clear recommendations for future improvements. It also proposes potential future enhancements for the tool, indicating a commitment to continuous development and improvement.

**Additional Feedback:**

no more questions.

**Correctness:**

Based on the provided excerpts from the submission, it appears that the authors' claims are well-supported by the evidence and analyses they provide.

**Documentation:**

yes

**Limitations:**

The authors should expedite their plans to incorporate sensor data into their tool. This would broaden its utility and relevance to the wider AV research community, especially those working in perception research.

**Opportunities For Improvement:**

Relevance to the Broader Research Community. The unified trajectory data loader doesn't yet incorporate sensor data, which restricts its application for perception research, an important area in AV research. Also, the current version of the tool doesn't support all available datasets, which limits its utility for researchers working with unsupported datasets.

Quality of the Research. The authors provide a comprehensive comparison of various datasets using "trajdata". However, they acknowledge that the lack of some feature comparisons (like geometric map element associations) could be seen as a limitation in their analysis.

**Relation To Prior Work:**

yes

**Summary And Contributions:**

This submission presents "trajdata," a unified interface that provides a uniform, simple, and efficient representation and API for various human trajectory datasets. The creation of trajdata is in response to the issue of numerous large-scale human trajectory datasets, used in autonomous vehicle and pedestrian motion tracking research, each using unique and custom data formats and APIs. This diversity often makes it challenging for researchers to train and evaluate methods across multiple datasets. By unifying these different datasets under a single API, trajdata aims to streamline the process, making it easier for researchers to compare and evaluate their methods. The paper also includes a comprehensive empirical evaluation of existing trajectory datasets, providing valuable insights into the current state of pedestrian and AV motion forecasting research, and offers suggestions for future datasets. The trajdata interface is freely available under the Apache 2.0 license.

---

> ### Author Response · Authors · 2023-08-11
> **Rebuttal for Reviewer 8txH**
>
> Thank you for your thoughtful review and suggestions! We agree that incorporating sensor data into trajdata and providing utilities to perform geometric map element associations would broaden its overall utility and relevance to the wider AV research community, and we are working towards both of these directions!

---

### Official Review · Reviewer_LYJj · 2023-07-21
**A Valuable Tool for Human/Vehicle Trajectory Prediction**

**Rating:** 7
**Confidence:** 4
**Clarity:** The exposition is very clear.

**Strengths:**

The authors have made it significantly easier to train forecasting models on 8 well-studied datasets, which will be appreciated by many researchers.
They have also provided helpful insights on how the datasets differ, which can aid in both debugging and analyzing prediction results on these datasets.

**Additional Feedback:**

N/A

**Correctness:**

The work seems technically sound and is further supported by the other works that have already leveraged this dataset.

**Documentation:**

The documentation for the dataset is sufficient and the authors provide code examples in the manuscript to illustrate the API's ease of use.
The authors have provided all required submission materials and details regarding collection, availability, and ethics.

**Ethics:**

I see no ethical issues with this work as the authors are simply harnessing publicly available benchmarks for human and vehicle trajectory prediction.

**Limitations:**

Within the scope of the manuscript, I cannot see any serious limitations. However, the authors were well-positioned to extend the data-loader to an evaluation testbed (e.g., closed-loop with nonreactive agents) and evaluation metrics for model predictions (e.g., ADE, heading error, collisions). While it is possible for others to create their own testbeds on top of the proposed dataloader, the standardization of an evaluation methodology would ensure that practitioners not only have access to the all of the same data but also the same evaluation method.

**Opportunities For Improvement:**

I think it would have been valuable to see other datasets included or explanations about why they were excluded (e.g., WILDTRACK and L-CAS).
It would also be good to see some discussion about simulated datasets like the ORCA-simulated subset of TrajNet++ or the Social-Force-simulated subset of A2X interaction benchmark, since hardly any real dataset demonstrates high-density emergency situations at a large scale. These are important, but very understudied scenarios that are prohibitively difficult to acquire (e.g., vehicle collisions).

**Relation To Prior Work:**

The connection between this work and prior work is clearly articulated; though it would be beneficial to have more insight into the datasets that were not chosen to be integrated, since prior multi-dataset benchmarks have included them (e.g., WILDTRACK and L-CAS).

**Summary And Contributions:**

The authors present a unified interface for various human and vehicle trajectory datasets, such as ETH/UCY, nuScenes, Waymo, and L5Kit (8 total datasets).
The authors also offer an evaluation of the datasets, comparing them across different metrics.

---

> ### Author Response · Authors · 2023-08-11
> **Rebuttal for Reviewer LYJj**
>
> Thank you for your thoughtful review and suggestions! We address detailed review comments point-by-point below.
>
> > other datasets included or explanations about why they were excluded (e.g., WILDTRACK and L-CAS)
>
> While building trajdata, we implemented support for datasets that are commonly used in the community and contain a variety of geographies, scenarios, and annotation schemes. There is no core methodological reason why certain datasets such as WILDTRACK and L-CAS are not yet supported. However, we do look forward to including these datasets and more within trajdata as our long-term goal is to support many more datasets and enable others to easily add support for new datasets. We have already made strides towards the latter by providing instructions for how to add support for new datasets in the README: https://github.com/NVlabs/trajdata#adding-new-datasets
>
> > It would also be good to see some discussion about simulated datasets like the ORCA-simulated subset of TrajNet++ or the Social-Force-simulated subset of A2X interaction benchmark
>
> When initially developing trajdata, we only planned to support human-generated motion data. However, as the reviewer states, hardly any real dataset demonstrates high-density emergency situations at a large scale. Thankfully, simulated datasets such as those in TrajNet++ or A2X can be added as easily as any other dataset since nothing in trajdata precludes the inclusion of synthetic data. Perhaps the only change required is to delineate which datasets are synthetic or not in [the supported datasets table](https://github.com/NVlabs/trajdata#supported-datasets) in the README when they are added, which we are happy to support.
>
> > However, the authors were well-positioned to extend the data-loader to an evaluation testbed (e.g., closed-loop with nonreactive agents) and evaluation metrics for model predictions (e.g., ADE, heading error, collisions).
>
> We agree, and we did! Lines 126-139 outline trajdata’s simulation interface, enabling users to control the motion of agents initialized from a specific timestep in a dataset.
>
> Due to the generality of the interface, trajdata enables evaluation in open-loop, closed-loop with nonreactive agents, and closed-loop with reactive agents settings (e.g., as in BITS [32] and CTG [33]). A full-fledged example is provided here: https://github.com/NVlabs/trajdata/blob/main/examples/sim_example.py In it, we show how a user can simulate a scene, measure and report simulation metrics (which trajdata provides in its [simulation module](https://github.com/NVlabs/trajdata/tree/main/src/trajdata/simulation), for full metric details see [sim_metrics.py](https://github.com/NVlabs/trajdata/blob/main/src/trajdata/simulation/sim_metrics.py) and [sim_stats.py](https://github.com/NVlabs/trajdata/blob/main/src/trajdata/simulation/sim_stats.py)), and save the resulting simulation trace for later analyses.

---

### Official Review · Reviewer_65pW · 2023-07-28

**Rating:** 6
**Confidence:** 3
**Clarity:** This paper is well-written and easy t…

**Strengths:**

- Multiple datasets are integrated into one unified format, enhancing diversity in use cases and training scenarios.

- A large volume of data has been compiled from 8 diverse datasets, including over three thousand data points and 200 million agents.

- Evaluations of existing datasets provide valuable insights for practitioners in this field.

**Additional Feedback:**

It would have been nicer to see some novel improvement of the existing algorithms based on deeper understanding about the multiple datasets to advance the technical level of this research field.

**Correctness:**

The contents discussed in this paper sound correct with proper data and explanation.

**Documentation:**

This paper seems to provide sufficient documentation.

**Ethics:**

No.

**Limitations:**

The authors did not state the limitations of this work.

**Opportunities For Improvement:**

- The overhead of employing the proposed APIs compared to using each dataset directly via its own API has not been addressed.

- There's no clear system in place for incorporating a new dataset into the existing collection.

- Despite the lack of new data collection or novel algorithm proposals, the insights offered are potentially beneficial to future research in human trajectory prediction.

**Relation To Prior Work:**

This paper seems to provide enough related work for background.

**Summary And Contributions:**

This paper proposed a unified interface for multiple human trajectory datasets called trajdata. The authors are motivated by the cumbersome research environment that requires handling various custom and unique data formats and APIs, which leads to the development of a unified interface for simple access to multiple popular human trajectory datasets. Using the proposed APIs, the authors further conducted comprehensive evaluations of the multiple datasets to provide insights to the researchers in this field. Although the proposed method and the provided insights would be useful and convenient, there seems to be no additional data collection or new algorithms to advance the research.

---

> ### Author Response · Authors · 2023-08-11
> **Rebuttal for Reviewer 65pW**
>
> Thank you for your thoughtful review and suggestions! We address detailed review comments point-by-point below.
>
> > The authors did not state the limitations of this work.
>
> The limitations of this work are discussed in Section 5 (Lines 301-302 and Lines 307-310) in the main text, and are referred to by Reviewer 8txH. We apologize for not highlighting them further.
>
> > The overhead of employing the proposed APIs compared to using each dataset directly via its own API has not been addressed.
>
> Thank you for raising this point. We have carefully designed trajdata to be efficient, parallelizable, and simple to use. [As illustrated in the README](https://github.com/NVlabs/trajdata#data-preprocessing-optional), trajdata employs a two-stage approach where the first stage formats and caches data into our unified format (only run once). The second stage then loads the data as per the user’s specifications in an efficient manner.
>
> This two-stage approach enables even more efficient parallel data loading than some dataset’s default APIs. As one concrete example, it is faster and less memory-intensive to load batches of nuScenes data in parallel with trajdata compared to its original devkit (which always loads the entire dataset into memory during initialization, quickly growing memory when replicated across multiple processes as is done during parallel dataloading).
>
> > There's no clear system in place for incorporating a new dataset into the existing collection.
>
> We do have a set of instructions for users wishing to incorporate a new dataset into trajdata, however, it is something we opted to outline in the repository’s README rather than the paper (due to the README’s closer association with trajdata’s code). Please find the instructions here: https://github.com/NVlabs/trajdata#adding-new-datasets

---

### Decision · Program_Chairs · 2023-09-22

**Decision:**

Accept (Poster)

**Comment:**

The authors have made training forecasting models on eight well-studied datasets significantly easier. The authors present a unified interface for various human and vehicle trajectory datasets, such as ETH/UCY, nuScenes, Waymo, and L5Kit (8 total datasets). The authors also evaluate the datasets, comparing them across different metrics.